# Leptin-Upregulated Metastasis-Associated Protein 1 Promotes Vasculogenic Mimicry in Breast Cancer Cells

**DOI:** 10.3390/ijms26125726

**Published:** 2025-06-15

**Authors:** Deok-Soo Han, Seung-Il Wang, Seung-Hyeon Lee, Eun-Ok Lee

**Affiliations:** 1Department of Science in Korean Medicine, College of Korean Medicine, Graduate School, Kyung Hee University, 26, Kyungheedae-ro, Dongdaemun-gu, Seoul 02447, Republic of Korea or hand2@uthscsa.edu (D.-S.H.); seung8543@khu.ac.kr (S.-I.W.); leesh3788@khu.ac.kr (S.-H.L.); 2Department of Center for Pain Therapeutics and Addiction Research, School of Dentistry, University of Texas Health Science Center at San Antonio, San Antonio, TX 78229, USA; 3Department of Cancer Preventive Material Development, College of Korean Medicine, Graduate School, Kyung Hee University, 26, Kyungheedae-ro, Dongdaemun-gu, Seoul 02447, Republic of Korea

**Keywords:** leptin, vasculogenic mimicry, breast cancer cells, metastasis-associated protein 1, Ob-R, STAT3

## Abstract

Leptin, a hormone primarily produced by adipose tissue, regulates energy balance and appetite, while contributing significantly to obesity and cancer progression. Vasculogenic mimicry (VM) refers to the process by which aggressive tumor cells form blood vessel-like structures, enabling blood supply independent of endothelial angiogenesis. Metastasis-associated protein 1 (MTA1) facilitates tumor progression and metastasis. This study investigated the role of MTA1 in the relationship between leptin and VM in human breast cancer cells. Leptin upregulated the mRNA and protein expression of MTA1, as revealed by a quantitative real-time PCR and Western blot analysis, respectively. However, the Western blot revealed that leptin-induced MTA1 upregulation was inhibited by the leptin receptor (Ob-R) blocker, Ob-R BP, and the signal transducer and activator of the transcription 3 (STAT3) inhibitor, AG490. The overexpression of MTA1 was observed to induce VM in a three-dimensional culture assay and to upregulate the expression of VM-related proteins, as confirmed by the Western blot. Conversely, silencing MTA1 suppressed leptin-induced VM and the expression of VM-related proteins. These findings indicate that leptin regulates MTA1 expression through the Ob-R/STAT3 signaling pathway and that MTA1 serves as a crucial mediator of leptin-induced VM.

## 1. Introduction

Vasculogenic mimicry (VM) serves as a critical blood supply mechanism, whereby aggressive tumor cells form vessel-like structures, independent of endothelial angiogenesis [1,2]. Unlike typical blood vessels formed by endothelial cells, VM blood vessels are lined with tumor cells that mimic endothelial cell functions [3]. This phenomenon is increasingly recognized as contributing to tumor progression, metastasis, and resistance to conventional treatments [4]. VM is particularly prevalent in highly aggressive cancers, such as breast cancer, and enhances the tumor’s ability to thrive and spread [5]. VM is associated with key molecular pathways and cellular processes, including the epithelial-to-mesenchymal transition (EMT), as well as the expression of specific matrix metalloproteinases and adhesion molecules [6,7]. Recent studies have identified multiple factors influencing VM, including growth factors, signaling molecules, and transcriptional regulators [8,9]. Elucidating these factors is essential for developing targeted therapies aimed at disrupting VM to improve cancer treatment outcomes [8,10]. The molecular mechanisms underlying VM are under active investigation to identify potential therapeutic strategies for its inhibition [11].

Leptin, a hormone primarily produced by adipose tissue, regulates energy balance and metabolism [12]. Beyond its established roles in appetite regulation and body weight control, leptin significantly influences cancer biology [13]. Recent studies have reported that leptin promotes tumor progression through mechanisms such as cell proliferation, survival, and angiogenesis [14,15]. Leptin has been shown to modulate tumor cell behavior, contributing to an aggressive phenotype in breast cancer [16]. In addition, leptin has been found to promote VM in breast cancer cells [17].

The metastasis-associated protein (MTA) family constitutes a critical component of the nuclear remodeling and deacetylation (NuRD) complex, which regulates target gene expression through the histone deacetylation of chromatin [18]. MTA1, a core member of the MTA family, is upregulated in numerous cancers and is associated with EMT, invasion, metastasis, angiogenesis, resistance to therapy, and poor prognosis in patients with cancer [19,20,21,22]. In breast cancer, MTA1 has been identified as a pivotal factor, enhancing tumor cell invasiveness and promoting metastasis [23,24]. Although extensive research has explored MTA1’s role in various cancer-related processes [25], its involvement in VM remains poorly understood.

This study aimed to clarify the mechanism by which leptin and MTA1 interact to induce VM in breast cancer cells, thereby providing new insights into the roles of leptin and MTA1 in tumor progression. Elucidating these interactions may facilitate the identification of new therapeutic targets for patients with breast cancer.

## 2. Results

### 2.1. Leptin Upregulates MTA1 Expression via the Ob-R/STAT3 Pathway in Human Breast Cancer Cells

To assess the effect of leptin on MTA1 expression, MDA-MB-231 and Hs 578T cells were treated with leptin for 24 h. Then, a quantitative real-time PCR was performed to measure the MTA1 mRNA expression, and Western blot analysis was conducted to evaluate the MTA1 protein expression. Leptin significantly upregulated the mRNA expression of MTA1 in both breast cancer cell lines (Figure 1A,B), indicating regulation at the transcriptional level. Similarly, the MTA1 protein expression was effectively upregulated by leptin treatment (Figure 1C,D). These results demonstrated that leptin regulates the expression of MTA1 in breast cancer cells.

To investigate the mechanism underlying leptin-induced MTA1 upregulation, the cells were treated with the leptin receptor blocker, Ob-R BP, and the STAT3 inhibitor, AG490, and their effects on MTA1 expression were assessed. The Ob-R BP treatment significantly attenuated the leptin-induced upregulation of MTA1 expression in both cell lines (Figure 2A,B), suggesting that leptin regulates MTA1 expression through its receptor. In addition, AG490 inhibited the leptin-induced MTA1 upregulation in a dose-dependent manner (Figure 2C,D), indicating that STAT3 signaling is essential for leptin-mediated MTA1 regulation. These results confirm that leptin activates the Ob-R/STAT3 signaling pathway to upregulate MTA1 expression in breast cancer cells.

### 2.2. MTA1 Overexpression Promotes VM in Human Breast Cancer Cells

To evaluate the role of MTA1 in VM in MDA-MB-231 and Hs 578T cells, a CRISPR activation plasmid was used. The Western blot confirmed that the MTA1 CRISPR activation plasmid effectively increased MTA1 expression compared to the control plasmid in both cell lines (Figure 3A,B). MTA1 overexpression significantly enhanced VM formation in both cell lines (Figure 3C,D). Furthermore, the expression of VM-related proteins, including VE-cadherin, Twist, MMP-2, and LAMC2, was upregulated by MTA1 overexpression in both cell lines (Figure 3E,F). These findings indicate that MTA1 plays a critical role in promoting VM in breast cancer cells by modulating the expression of VM-related proteins.

### 2.3. MTA1 Silencing Inhibits Leptin-Induced VM in Human Breast Cancer Cells

To examine the role of MTA1 in leptin-induced VM, MDA-MB-231 and Hs 578T cells were transfected with MTA1 siRNA for 48 h and, subsequently, treated with leptin for 24 h. The Western blot demonstrated that the MTA1 siRNA effectively reduced the leptin-induced upregulation of MTA1 expression compared to the control siRNA in both cell lines (Figure 4A,B). MTA1 silencing markedly suppressed leptin-induced VM formation in both cell lines (Figure 4C,D). Moreover, the leptin-induced upregulation of VM-related proteins, including VE-cadherin, Twist, MMP-2, and LAMC2, was attenuated by MTA1 silencing in both cell lines (Figure 4E,F). These findings confirm that MTA1 mediates leptin-induced VM in breast cancer cells.

## 3. Discussion

VM refers to the process by which aggressive tumor cells form blood vessel-like structures to supply blood, thereby promoting tumor growth and metastasis [26]. VM is associated with poor prognosis, increased aggressiveness, and metastasis in various cancers, including breast cancer. Notably, VM contributes to drug resistance, which may account for the limited efficacy of antiangiogenic therapies in breast cancer [27]. Elucidating the molecular mechanisms underlying VM is critical for developing novel therapeutic strategies for breast cancer. 

Leptin, an adipokine secreted by adipose tissue, is implicated in breast cancer development and represents a potential therapeutic target, particularly for obese women with breast cancer [28]. Leptin activates Wnt1 signaling by upregulating MTA1 expression, which induces EMT in breast cancer cells [29]. As a transcriptional co-regulator, MTA1 is involved in tumor metastasis and progression [22]. These findings suggest that MTA1 is a novel target of leptin and plays a key role in mediating its biological effects. Consistent with these findings, the present study demonstrated that leptin significantly upregulated MTA1 expression at both the mRNA and protein levels (Figure 1). EMT, a process by which epithelial cells acquire mesenchymal characteristics with enhanced motility and invasiveness, is essential for tumor progression and metastasis. EMT-related transcription factors are upregulated in VM-forming tumor cells, and regulators of EMT promote VM by inducing this transition [30,31]. A previous study showed that leptin induces VM in breast cancer cells [17]. Consequently, leptin is hypothesized to promote VM in breast cancer cells by interacting with MTA1. This study aimed to clarify the mechanisms by which leptin and MTA1 interact to drive VM in breast cancer cells.

The qPCR and Western blot analysis revealed a positive correlation between MTA1 expression and leptin treatment in both breast cancer cell lines (Figure 1). Leptin binds to specific leptin receptors (Ob-Rs), leading to the activation of various intracellular signaling pathways, including STAT3 signaling [32,33]. The leptin/Ob-R/STAT3 signaling axis plays a critical role in breast cancer development by regulating target genes involved in tumor growth and progression [34,35]. To determine whether the leptin receptor and STAT3 signaling are involved in the leptin-induced upregulation of MTA1 expression, both breast cancer cell lines were treated with a leptin receptor blocker (Ob-R BP) and a STAT3 inhibitor (AG490). The Western blot results showed that leptin-induced MTA1 upregulation was suppressed by all these inhibitors (Figure 2), confirming that leptin regulates MTA1 expression via the Ob-R/STAT3 signaling pathway in breast cancer cells. These findings suggest that MTA1 may be a target gene of the leptin/Ob-R/STAT3 signaling pathway.

The novel role of MTA1 in VM was confirmed through gain-of-function and loss-of-function studies. The overexpression of MTA1, achieved using the CRISPR activation plasmid, significantly promoted VM formation (Figure 3A–D), indicating that MTA1 is pivotal in facilitating VM in breast cancer cells. Conversely, the reduction in MTA1 caused by the MTA1 siRNA suppressed leptin-induced VM (Figure 4A–D). As a regulator of the chromatin structure, MTA1 interacts with transcription factors to modulate gene expression associated with aggressive cancer phenotypes [36,37]. To determine whether MTA1 regulates VM-related protein expression, Western blot analysis was performed. Increased MTA1 expression was found to upregulate vascular endothelial cadherin (VE-cadherin), a marker of endothelial cell junctions; Twist, a transcription factor promoting EMT and regulating VE-cadherin expression; matrix metalloproteinase-2 (MMP-2), a gelatinase that degrades and remodels the extracellular matrix (ECM); and laminin subunit 5 gamma-2 (LAMC2), a component of laminin-5 involved in cell adhesion and basement membrane formation (Figure 3E,F) [38,39]. Conversely, reduced MTA1 expression resulted in decreased levels of VM-related proteins upregulated by leptin (Figure 4E,F). These findings indicate that MTA1 promotes VM by modulating the expression of proteins involved in cell adhesion, migration, invasion, and ECM remodeling.

In conclusion, leptin was found to upregulate MTA1 expression via the Ob-R/STAT3 signaling pathway, which in turn promotes VM by regulating the expression of VM-related proteins in breast cancer cells. These findings highlight the critical role of MTA1 in VM regulation and suggest its potential as a therapeutic target. Inhibiting MTA1 may suppress VM, thereby reducing tumor progression and metastasis. However, further studies are required to elucidate the precise mechanisms by which MTA1 contributes to tumor progression.

## 4. Materials and Methods

### 4.1. Cell Culture

Human triple-negative breast cancer cell lines, MDA-MB-231 and Hs 578T, were obtained from the Korean Cell Line Bank (Seoul, Republic of Korea). RPMI1640 (Cat: LM 011-01, WELGENE, Daegu, Republic of Korea) and DMEM (Cat: LM 001-05, WELGENE), supplemented with 10% fetal bovine serum (Cat: S101-07, WELGENE) and 1% antibiotics (Cat: LS203-01, WELGENE), were used to culture the MDA-MB-231 and Hs 578T cells, respectively. Both cell lines were maintained in a humidified incubator at 37 °C, with 5% CO_2_.

### 4.2. RNA Isolation and Quantitative Real-Time PCR

Total RNA was extracted from the cells using TRIzol reagent (Invitrogen, Carlsbad, CA, USA), according to the manufacturer’s protocol. Complementary DNA (cDNA) was synthesized from 2 µg of total RNA, using M-MLV reverse transcriptase (Promega, Madison, WI, USA). A quantitative real-time PCR (qPCR) was performed, using GreenStar™ qPCR Master Mix (Bioneer Corporation, Daejeon, Republic of Korea), with specific primers (Table 1), using the Thermal Cycler Dice^®^ Real Time System III (Takara, Katsushika City, Tokyo, Japan).

### 4.3. Western Blot Analysis

Cell lysates were subjected to SDS-PAGE (8–12% gels) at a constant voltage of 60–85 V and transferred to membranes (Pall Corporation, Port Washington, NY, USA) at 100 V for 70–90 min. Membranes were blocked with 5% skim milk or bovine serum albumin for 90 min, then incubated with specific primary antibodies (Table 2) at 4 °C for 24 h, followed by incubation with appropriate secondary antibodies (Table 2) at room temperature for 2 h. Protein bands were detected using an enhanced chemiluminescent reagent (GE Healthcare, Chicago, IL, USA).

### 4.4. MTA1 Overexpression Using CRISPR Activation Plasmid

MTA1 was overexpressed using a CRISPR activation system. MDA-MB-231 cells (1.5 × 10^5^) and Hs 578T cells (2.0 × 10^5^) were cultured under optimal conditions and transfected with 1 μg of the control or 0.5 μg of the MTA1 CRISPR activation plasmid (Santa Cruz, Danvers, MA, USA) for 48 h. Transfection was performed using the UltraCruz transfection reagent (Santa Cruz), according to the manufacturer’s protocol.

### 4.5. MTA1 Silencing Using Small Interfering RNA

MTA1 was silenced using a small interfering RNA (siRNA). MDA-MB-231 (1.5 × 10^5^) and Hs 578T (2.0 × 10^5^) cells were cultured under optimal conditions and transfected with 5 nM of the control or the MTA1 siRNA (Santa Cruz) for 48 h. Transfection was performed using the INTERFERin transfection reagent (Polyplus Transfection, New York, NY, USA), according to the manufacturer’s protocol.

### 4.6. Three-Dimensional (3D) Culture VM Tube Formation Assay

To evaluate the formation of blood vessel-like structures by tumor cells, 24-well plates were coated with 100 μL of Matrigel, followed by a 3D culture assay conducted for 16 h [17,38]. To assess the gain-of-function of MTA1, MDA-MB-231 cells (2.5 × 10^5^) and Hs 578T cells (3.5 × 10^5^) were transfected with CRISPR activation plasmids (Santa Cruz Biotechnology). For the loss-of-function studies, both cell lines were treated with leptin after transfection with siRNA (Santa Cruz Biotechnology). The formed VM structures were visualized and quantified using a Ts2_PH optical microscope (Nikon, Tokyo, Japan) at 40× magnification.

### 4.7. Statistical Analysis

The results are expressed as mean ± standard deviation (SD) from three independent experiments. Statistical significance was determined using GraphPad Prism software (version 5, GraphPad Software Inc., Boston, MA, USA), with Student’s *t*-test (*p* < 0.05).

## Figures and Tables

**Figure 1 ijms-26-05726-f001:**
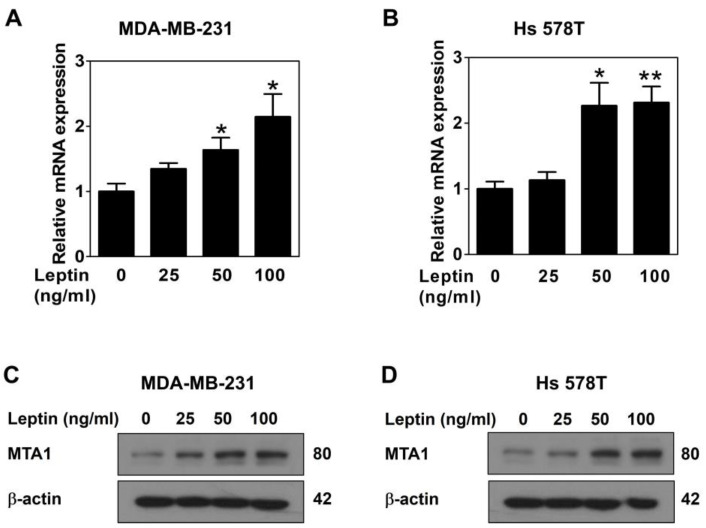
Leptin upregulates MTA1 expression in human breast cancer cells. MTA1 mRNA expression was analyzed using qPCR in MDA-MB-231 (**A**) and Hs 578T cells (**B**) after leptin treatment for 24 h. Results are expressed as mean ± SD; * *p* < 0.05 and ** *p* < 0.01 vs. untreated control. MTA1 protein expression was assessed using Western blot in MDA-MB-231 (**C**) and Hs 578T cells (**D**) after 24 h of leptin treatment.

**Figure 2 ijms-26-05726-f002:**
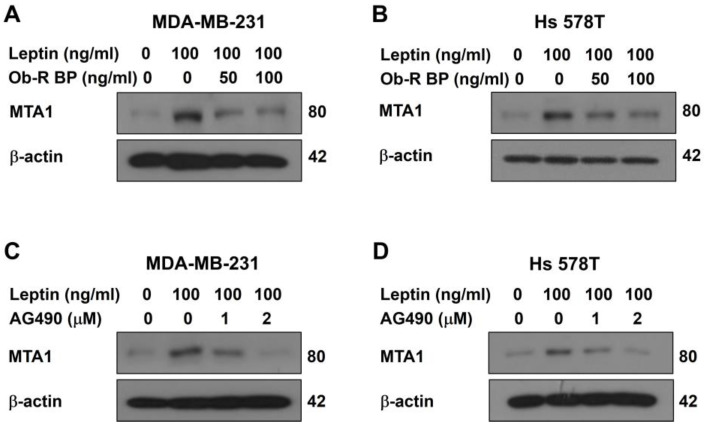
Leptin upregulates MTA1 expression via the Ob-R/STAT3 pathway in human breast cancer cells. MTA1 protein expression was evaluated using Western blot in MDA-MB-231 (**A**,**C**) and Hs 578T cells (**B**,**D**) after 24 h of leptin treatment in the presence or absence of Ob-R BP (**A**,**B**) or AG490 (**C**,**D**).

**Figure 3 ijms-26-05726-f003:**
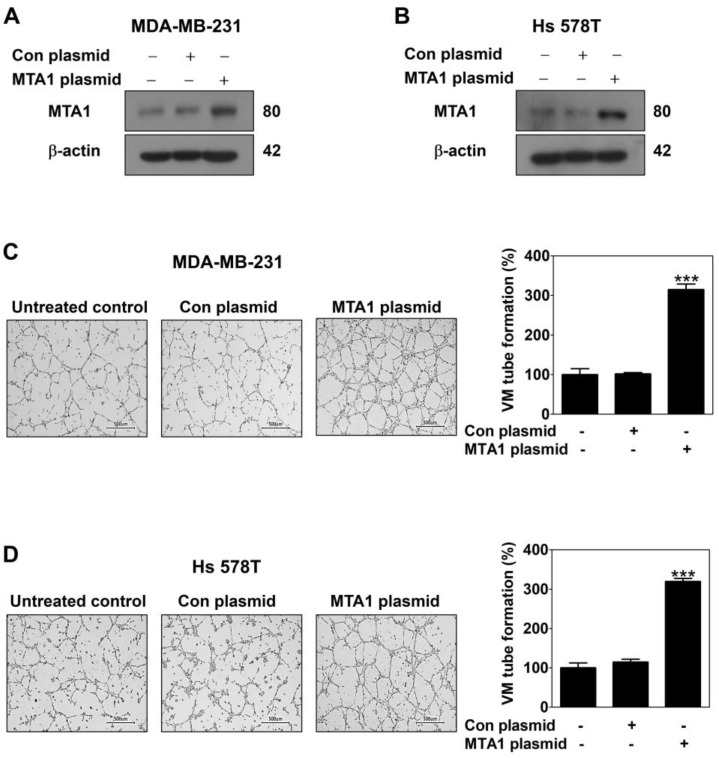
MTA1 overexpression promotes VM in human breast cancer cells. MDA-MB-231 (**A**,**C**,**E**) and Hs 578T cells (**B**,**D**,**F**) were transfected with the MTA1 CRISPR activation plasmid for 48 h. MTA1 protein expression was assessed using Western blot in MDA-MB-231 (**A**) and Hs 578T cells (**B**). VM was performed using a 3D culture assay for 16 h in MDA-MB-231 (**C**) and Hs 578T cells (**D**) (40× magnification; scale bar = 500 μm). Results are expressed as mean ± SD; *** *p* < 0.001 vs. control plasmid. Expression of VM-related proteins was evaluated using Western blot in MDA-MB-231 (**E**) and Hs 578T cells (**F**).

**Figure 4 ijms-26-05726-f004:**
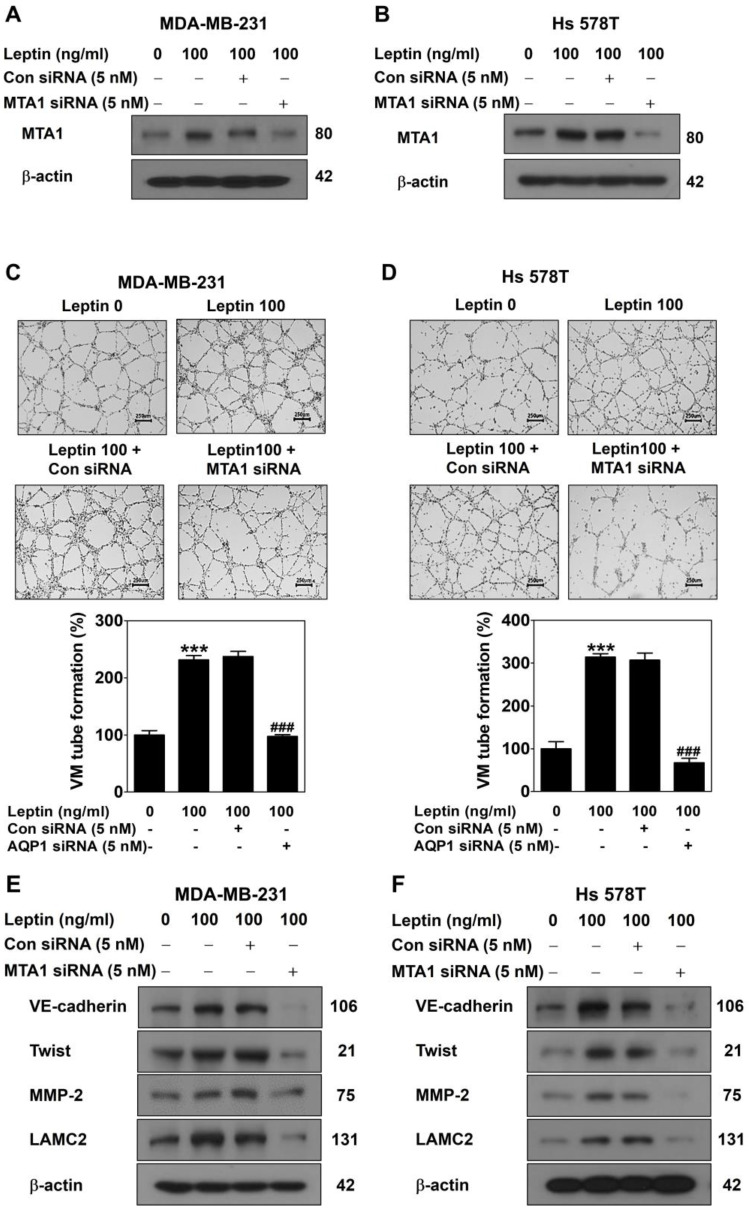
MTA1 silencing inhibits leptin-induced VM in human breast cancer cells. MDA-MB-231 (**A**,**C**,**E**) and Hs 578T cells (**B**,**D**,**F**) were treated with leptin 48 h after transfection with MTA1 siRNA. MTA1 protein expression was evaluated using Western blot in MDA-MB-231 (**A**) and Hs 578T cells (**B**). VM was performed using a 3D culture assay for 16 h in MDA-MB-231 (**C**) and Hs 578T cells (**D**) (40× magnification; scale bar = 250 μm). Results are expressed as mean ± SD; *** *p* < 0.001 vs. untreated control; ### *p* < 0.001 vs. control siRNA. Expression of VM-related proteins was evaluated using Western blot in MDA-MB-231 (**E**) and Hs 578T cells (**F**).

**Table 1 ijms-26-05726-t001:** Primers used in this study.

mRNA	Primer Sequences
β-actin	S: 5′-AAGAGAGGCATCCTCACCCT-3′ AS: 5′-ATCTCTTGCTCGAAGTCCAG-3′
MTA1	S: 5′-CCAGGACCAAACCGCAGTAACA-3′ AS: 5′-GTCAGCTTCGTCGTGTGCAGAT-3′

**Table 2 ijms-26-05726-t002:** Antibodies used in this study.

Antibody	Company	Dilution	Product No.
MTA1	Santa Cruz	1:500	SC-17773
β-actin	Sigma-Aldrich	1:20,000	A5316
p-STAT3	CST	1:1000	9145
STAT3	CST	1:5000	12640
MMP-2	Abcam	1:1000	ab86607
LAMC2	Abcam	1:1000	ab96327
VE-cadherin	Abgent	1:1000	AP2724a
Twist	Abcam	1:1000	ab50887
Goat anti-rabbit IgG-HRP	CST	1:5000	7074P2
Goat anti-mouse IgG-HRP	Bio-Rad	1:5000	STAR120P

Santa Cruz (Danvers, MA, USA); CST, Cell Signaling Technology (Beverly, MA, USA); Sigma-Aldrich (St. Louis, MO, USA); Abcam (Cambridge, UK); Abgent (San Diego, CA, USA); Bio-Rad (Hercules, CA, USA).

## Data Availability

The data is contained within the article.

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
