# Peer review of "Leptin-Upregulated Metastasis-Associated Protein 1 Promotes Vasculogenic Mimicry in Breast Cancer Cells"

_ijms, 2025, doi:10.3390/ijms26125726_

Round 1

Reviewer 1 Report

Comments and Suggestions for Authors

This article reports the role of MTA1 in the relationship between leptin and VM in human breast cancer cells. The novelty of this study was to provide new insights into the molecular mechanisms by which leptin promotes vasculogenic mimicry in cancer. Specifically, this work identified metastasis-associated protein 1 as a novel downstream effector of leptin signaling. However, there are several major concerns as follows:

  1. There are so many typos. The manuscript is suggested for English language corrections and improvements.
  2. Results:
  • Figure 1: The authors should clarify the meaning of the asterisks (* and **) used in the figure. Typically, these symbols indicate levels of statistical significance, and their corresponding p-values should be clearly defined in the figure legend.
  • Figures 1-4: The molecular weight of each protein should be indicated in the Western blot data to enhance interpretability and confirm protein identity.
  • The authors should justify why normal human breast cells were not included as a control group for comparison with breast cancer cells. Including such controls would strengthen the conclusions by providing a baseline reference.
  • Figure 3: The terms “Control” and “Control plasmid” should be clearly defined and consistently used in the figure and figure legend to avoid confusion.
  • Figures 3C and 4C: The image quality is suboptimal. Please provide higher-resolution images to ensure the data are clearly visible and appropriately interpretable.
  1. Please cross check the references in list of references and citations in the text.
Comments on the Quality of English Language

The English could be improved to more clearly express the research.

Author Response

Comment 1: There are so many typos. The manuscript is suggested for English language corrections and improvements.

Response 1: We have already completed the English corrections.

Comment 2: Figure 1: The authors should clarify the meaning of the asterisks (* and **) used in the figure. Typically, these symbols indicate levels of statistical significance, and their corresponding p-values should be clearly defined in the figure legend.

Response 2: According to your comment, we have added statistical explanations.

Comment 3: Figures 1-4: The molecular weight of each protein should be indicated in the Western blot data to enhance interpretability and confirm protein identity.

Response3: According to your comment, we have indicated the molecular weight.

Comment 4: The authors should justify why normal human breast cells were not included as a control group for comparison with breast cancer cells. Including such controls would strengthen the conclusions by providing a baseline reference.

Response 4: Normal cells were excluded from experiments because they cannot form VM.

Comment 5: Figure 3: The terms “Control” and “Control plasmid” should be clearly defined and consistently used in the figure and figure legend to avoid confusion.

Response 5: According to your comment, we have corrected it.

Comment 6: Figures 3C and 4C: The image quality is suboptimal. Please provide higher-resolution images to ensure the data are clearly visible and appropriately interpretable.

Response 6: According to your comment, we have corrected it.

Comment 7: Please cross check the references in list of references and citations in the text.

Response 7: We have corrected it.

Reviewer 2 Report

Comments and Suggestions for Authors

This manuscript offers valuable information to the cell signaling literature through the leptin MTA-1 pathway. Information from this study should be published as soon as possible to allow other researchers to proceed with the next steps in cancer research. 

  1. The introduction/background section provides an excellent set-up for the paper, preparing the reader for the study.
  2. The results section is well developed and explained, providing specific details to allow for replication of the study. 
  3. The discussion section gives adequate information about how this study relates to other studies and provides direction for future studies. 
  4. The methods section presents the specific information for study replication. 
  5. The reference list is appropriate; however, it seems that citation numbers 38 and 40 are the same, differing only in capitalization, with one having a page number listed. 

In my opinion, no significant changes are required for publication. 

Author Response

Comment 1: The introduction/background section provides an excellent set-up for the paper, preparing the reader for the study.

Response 1: Thank you for your valuable comments.

Comment 2: The results section is well developed and explained, providing specific details to allow for replication of the study. 

Response 2: Thank you for your valuable comments.

Comment 3: The discussion section gives adequate information about how this study relates to other studies and provides direction for future studies. 

Response 3: Thank you for your valuable comments.

Comment 4: The methods section presents the specific information for study replication. 

Response 4: Thank you for your valuable comments.

Comment 5: The reference list is appropriate; however, it seems that citation numbers 38 and 40 are the same, differing only in capitalization, with one having a page number listed. 

Response 5: We have corrected it.

Reviewer 3 Report

Comments and Suggestions for Authors

In this manuscript, the authors present compelling evidence that leptin upregulates MTA1 expression in breast cancer cells. They further demonstrate that MTA1 contributes to vasculogenic mimicry (VM) in these cells, and that silencing MTA1 effectively inhibits leptin-induced VM formation. Overall, the manuscript is clearly written, and the experimental design and execution are of high quality.

However, I have the following concerns and suggestions that should be addressed to strengthen the study:

  1. Did the authors explore publicly available datasets (e.g., TCGA) to assess MTA1 expression levels across different breast cancer molecular subtypes? Such an analysis could provide valuable context regarding the clinical relevance of MTA1.

  2. Were any pharmacological inhibitors of MTA1 tested in this study? If not, the inclusion of such data could provide insight into the potential translational applications of targeting MTA1.

  3. Does inhibition of MTA1 sensitize breast cancer cells to standard anticancer therapies (e.g., chemotherapy, targeted therapy)? Exploring this could uncover synergistic treatment strategies.

  4. Were in vivo experiments conducted involving MTA1 overexpression or inhibition? Animal studies would significantly enhance the translational impact of the findings.

Addressing these points will further validate the conclusions and increase the clinical relevance of the study.

Author Response

Comment 1: Did the authors explore publicly available datasets (e.g., TCGA) to assess MTA1 expression levels across different breast cancer molecular subtypes? Such an analysis could provide valuable context regarding the clinical relevance of MTA1.

Response 1: Thank you for your valuable opinion. While not yet attempted in this study, this area would likely need to be explored to further elucidate the function of MTA1.

Comment 2: Were any pharmacological inhibitors of MTA1 tested in this study? If not, the inclusion of such data could provide insight into the potential translational applications of targeting MTA1.

Response 2: No. We were unable to purchase commercially available inhibitors.

Comment 3 : Does inhibition of MTA1 sensitize breast cancer cells to standard anticancer therapies (e.g., chemotherapy, targeted therapy)? Exploring this could uncover synergistic treatment strategies.

Response 3: Thank you for your valuable opinion. We think this area needs more research.

Comment 4: Were in vivo experiments conducted involving MTA1 overexpression or inhibition? Animal studies would significantly enhance the translational impact of the findings. Addressing these points will further validate the conclusions and increase the clinical relevance of the study.

Response 4: No, but after deriving a drug that inhibits MTA1 expression, we plan to verify its effectiveness through animal experiments. Thank you for your valuable opinion.

Reviewer 4 Report

Comments and Suggestions for Authors

In this manuscript, the authors assessed the role of leptin in upregulating MTA1 expression and attempted to elucidate the mechanism of interaction between leptin and MTA1 to induce vasculogenic mimicry (VM) in breast cancer cells. This study will definitely contribute to better understanding of the role of MTA1 in VM.  I therefore recommend acceptance after the following points have been addressed:

  1. The study used to cancer cell line models. Why didn't the authors include a healthy cell line as potential negative control?
  2. Fig. 2 shows that leptin upregulates MTA1 expression via the Ob-R/STAT3 pathway in human breast cancer cells. Could other pathways be possible?

Author Response

Comment 1: The study used to cancer cell line models. Why didn't the authors include a healthy cell line as potential negative control?

Response 1: Normal cells were excluded from experiments because they cannot form VM.

Comment 2: Fig. 2 shows that leptin upregulates MTA1 expression via the Ob-R/STAT3 pathway in human breast cancer cells. Could other pathways be possible?

Response 2: We expect that they could be possible.

Round 2

Reviewer 1 Report

Comments and Suggestions for Authors

The authors answered the questions reasonably.